# Memory strength gates the involvement of a CREB-dependent cortical fear engram in remote memory

Mariana R. Matos[1], Esther Visser [1], Ioannis Kramvis[1], Rolinka J. van der Loo[1], Titia Gebuis[1], Robbert Zalm[2], Priyanka Rao-Ruiz [1], Huibert D. Mansvelder [3], August B. Smit[1] & Michel C. van den Oever [1]

Encoding and retrieval of contextual memories is initially mediated by sparsely activated neurons, so-called engram cells, in the hippocampus. Subsequent memory persistence is thought to depend on network-wide changes involving progressive contribution of cortical regions, a process referred to as systems consolidation. Using a viral-based TRAP (targeted recombination in activated populations) approach, we studied whether consolidation of contextual fear memory by neurons in the medial prefrontal cortex (mPFC) is modulated by memory strength and CREB function. We demonstrate that activity of a small subset of mPFC neurons is sufficient and necessary for remote memory expression, but their involvement depends on the strength of conditioning. Furthermore, selective disruption of CREB function in mPFC engram cells after mild conditioning impairs remote memory expression. Together, our data demonstrate that memory consolidation by mPFC engram cells requires CREB-mediated transcription, with the functionality of this network hub being gated by memory strength.

---

[1] Department of Molecular and Cellular Neurobiology, Center for Neurogenomics and Cognitive Research, Amsterdam Neuroscience, Vrije Universiteit Amsterdam, 1081 HV Amsterdam, The Netherlands. [2] Department of Functional Genomics, Center for Neurogenomics and Cognitive Research, Amsterdam Neuroscience, Vrije Universiteit Amsterdam, 1081 HV Amsterdam, The Netherlands. [3] Department of Integrative Neurophysiology, Center for Neurogenomics and Cognitive Research, Amsterdam Neuroscience, Vrije Universiteit Amsterdam, 1081 HV Amsterdam, The Netherlands. Correspondence and requests for materials should be addressed to M.O. (email: michel.vanden.oever@vu.nl)

n recent years, great progress has been made in understanding the neurobiological substrate, or engram, of a recently acquired (<1-week-old) memory. For instance, initial formation and expression of conditioned-fear memory is mediated by coordinated activity of small subsets of neurons, referred to as neuronal ensembles[1] or engram cells[2], in hippocampal circuitry and the amygdala[3–6]. Persistence of memory is thought to depend on systems consolidation, a time-dependent process through which a given memory is gradually consolidated in cortical networks[7,8]. This concept is supported by the observation that retrieval of contextual fear memory initially does not depend on activity in cortical areas, including the mPFC, however, cortical activity is required for memory retrieval at remote timepoints[9–11]. Despite these findings, the molecular and cellular mechanisms that support the consolidation of remote (≥month-old) memories and the influence of memory strength on the engagement of cortical neuronal ensembles in remote memory expression are poorly understood. This is mainly because of

technical limitations to selectively manipulate subsets of neurons several weeks after they are activated by a specific experience.

Memory consolidation initially depends on de novo RNA and protein synthesis[12], critical for changes in structural plasticity and synaptic strength between neurons in an engram network[13]. It is well-established that the transcription factor CREB (cAMP-responsive element binding protein) has a crucial role in regulating gene expression that underlies formation of long-term memory[14], as determined by systemic knock-out or global knock-down of CREB function[15–17]. More recently, it was shown that modulation of CREB function in small subsets of neurons prior to learning affects the probability that these cells will be incorporated in a memory engram[18,19], suggesting that differences in CREB levels at the time of learning determine which neurons will become engram cells[20]. However, whether CREB function in cortical engram cells is required after learning to support systems consolidation of memory has never been demonstrated.

Here, we investigated (1) whether the mPFC harbors engram cells supporting remote contextual fear memory; (2) whether involvement of mPFC neurons is modulated by the strength of conditioning; and (3) whether CREB function in these neurons is required for memory persistence. To test this, we developed a dual-virus variant of TRAP[21]. This allowed us to express a lasting molecular tag (e.g. Designer Receptor Exclusively Activated by Designer Drugs (DREADD)[22] or mCREB (a repressor of CREB function)) in activated neurons of wild-type mice. Using this system, we found that engram cells in the mPFC are already defined during learning, but their functional contribution to memory expression requires CREB-mediated transcription and depends on memory strength.

## Results

**Fear conditioning evokes neuronal activation in the mPFC**. We first assessed whether neurons in the dorsal mPFC region (comprising the prelimbic cortex (PL) and anterior cingulate cortex (ACC)) were activated during CFC (using a single foot-shock = unconditioned stimulus (US)) or exposure to the CFC context in absence of a foot-shock (Ctx). Mice that remained in their home-cage were used as controls (HC; Fig. 1a). An activated neuron was defined by the expression of the immediate early gene Fos (Fos expression is rapidly and transiently induced by neuronal activity;[23] Fig. 1b). Compared with HC controls (1.7 ± 0.1%; mean±SEM Fos⁺ neurons), the percentage of Fos⁺ neurons was significantly enhanced in Ctx (7.3 ± 0.1%) and CFC (8.5 ± 0.4%) groups (Fig. 1c). A minor, but significant, difference was found between Ctx and CFC mice. Hence, sparse neuronal activity is induced in the mPFC by CFC, as well as by mere exposure to a novel context. Therefore, we next investigated the functional relevance of CFC-activated mPFC ensembles.

**Viral-TRAP enables molecular tagging of activated neurons**. To enable chemogenetic manipulation of CFC-activated mPFC neurons during retrieval of a recent (<1-week-old) and remote (1-month-old) fear memory, we developed an inducible dual-virus system based on TRAP[21]. This method comprised an Adeno-Associated Virus (AAV) coding for inducible Cre recombinase under the control of the Fos promoter (AAV-Fos:: CreER^T2) and a second Cre-dependent AAV, e.g., containing the coding sequence of hM4Di (inhibitory Gi-DREADD[24]) in an inverse open reading frame and flanked by Cre recognition sites. With this method, CreER^T2-mediated hM4Di expression is coupled to the Fos promoter and controlled by systemic injection of 4-hydroxytamoxifen (4TM;[21,25] Fig. 2a). Mice received an AAV mixture of Fos::CreER^T2 and Cre-dependent hM4Di fused to mCherry (fluorescent reporter) in the dorsal mPFC and remained

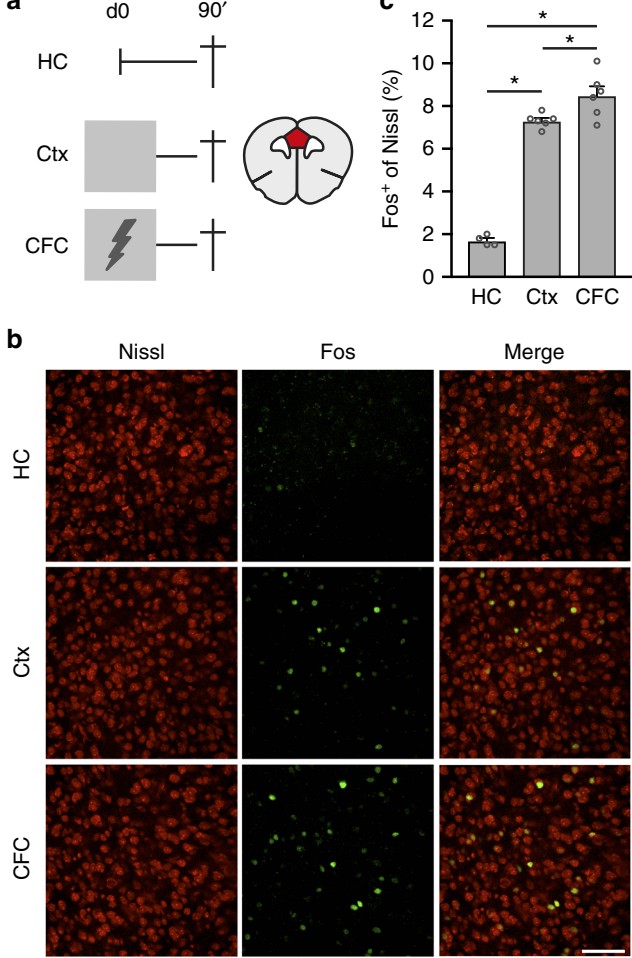

**Fig. 1** CFC enhanced neuronal activity in the mPFC. **a** Left: experimental design of groups used to assess Fos expression. Home-cage (HC; n = 4), context exposure only (Ctx; n = 6), contextual fear conditioning (CFC; n = 6). Lightning bold indicates foot-shock (1US). Right: illustration of a coronal brain section indicating the mPFC region (red) where Fos⁺ neurons were analyzed. **b** Representative examples of Fos⁺ cells (green) in all groups. **c** Percentage of Fos⁺ cells in each group. One-way ANOVA $F_{(2,13)} = 126.3$, $p < 0.0001$. Post-hoc Bonferroni test: HC vs. Ctx *$p < 0.0001$, HC vs. CFC *$p < 0.0001$, Ctx vs. CFC *$p = 0.032$. Scale bar = 50 µm. Bar graph shows mean + s.e.m. Source data are provided as a Source Data file

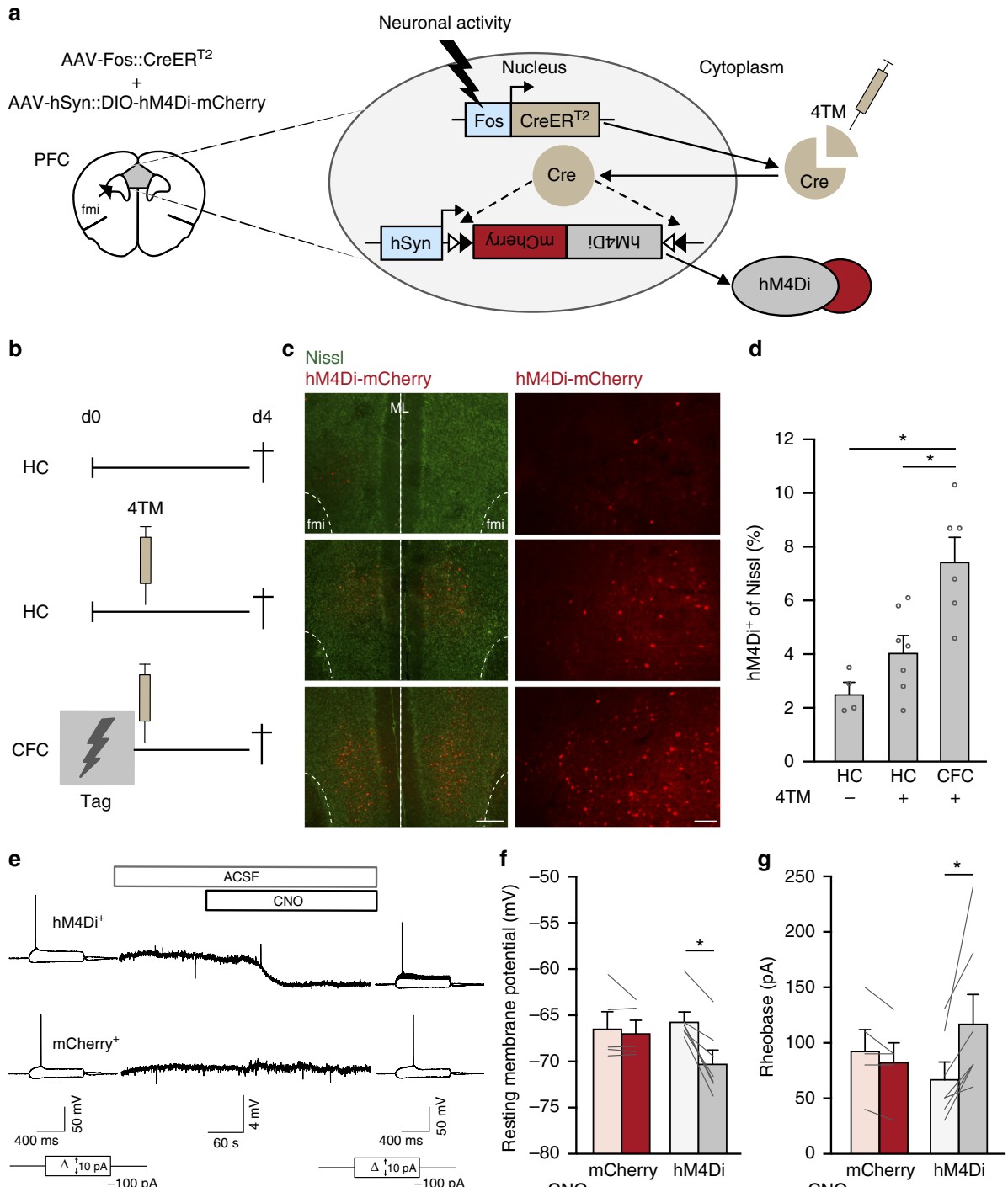

**Fig. 2** Viral-TRAP enables inducible activity-dependent tagging of mPFC neurons. **a** Schematic representation of the viral-TRAP method. A mixture of AAV-Fos::CreER$^{T2}$ and Cre-dependent AAV (e.g. AAV-hSyn::DIO-hM4Di-mCherry) is bilaterally infused into the mPFC. The *Fos* promoter is activated by neuronal activity, resulting in CreER$^{T2}$ expression. Systemic injection of 4-hydroxytamoxifen (4TM) allows translocation of CreER$^{T2}$ into the nucleus enabling irreversible recombination of the Cre-dependent vector and expression of hM4Di-mCherry driven by the human *Synapsin* (hSyn) promoter. fmi = forceps minor of the corpus callosum. **b** Experimental design of groups used to validate viral-TRAP. Home-cage (HC) −4TM ($n = 4$), HC + 4TM ($n = 7$), Contextual fear conditioning (CFC) + 4TM ($n = 6$). 4TM was injected systemically 2 h after CFC on day 0 and all groups were killed 4 days later. **c** Expression of hM4Di-mCherry in mPFC. fmi = forceps minor of the corpus callosum. ML = midline. Left: scale bar = 250 μm; Right: scale bar = 100 μm. **d** Percentage of hM4Di-mCherry$^+$ cells in mPFC. One-way ANOVA: $F_{(2,14)} = 12.3$, $p = 0.001$; post-hoc Bonferroni test: CFC vs. HC −4TM, $p = 0.001$, CFC vs. HC + 4TM, *$p = 0.007$. **e** Patch-clamp recordings of hM4Di$^+$ and mCherry$^+$ (control) cells before and after CNO application. ACSF = artificial cerebrospinal fluid. **f** Resting membrane potential changes for mCherry$^+$ ($n = 5$) and hM4Di$^+$ ($n = 7$) neurons. Wilcoxon signed rank test, hM4Di$^+$: $Z = −2.37$, *$p = 0.018$; mCherry$^+$: $Z = −0.41$ $p = 0.69$. **g** Rheobase changes for mCherry$^+$ ($n = 5$) and hM4Di$^+$ ($n = 7$) neurons. Wilcoxon signed rank test, hM4Di$^+$: $Z = −2.38$ *$p = 0.018$; mCherry$^+$: $Z = −1.63$, $p = 0.10$. All bar graphs show means + s.e.m. Source data are provided as a Source Data file

in their home-cage or underwent CFC followed by 4TM treatment (Fig. 2b). CFC + 4TM induced hM4Di-mCherry expression in 7.5 ± 0.9% (mean±SEM) of mPFC neurons (Fig. 2c, d; Supplementary Fig. 1), similar to the percentage of Fos+ cells induced by CFC (Fig. 1c). Home-cage control mice without (HC −4TM) and with (HC + 4TM) 4TM treatment showed significantly less hM4Di-expressing (hM4Di+) neurons (2.5 ± 0.4% and 4.1 ± 0.6%, respectively), confirming that this technique enabled activity-dependent tagging of mPFC neurons. In addition to a difference in the number of tagged cells, the fluorescence intensity of hM4Di+ cells seemed higher in 4TM treated groups (Fig. 2c). Next, we assessed functionality of hM4Di expression by patch-clamp recordings in acute brain slices 4–7 days after CFC. Clozapine N-oxide (CNO) reduced the resting membrane potential and increased the depolarization threshold (rheobase) in hM4Di+ neurons, but not in neurons that expressed mCherry alone (Fig. 2e–g). This indicates that CNO reduced excitability of hM4Di+ mPFC neurons, enabling suppression of their activity.

**CFC-tagged mPFC neurons are required for remote memory**. To determine whether mPFC neurons activated during CFC are involved in recent and remote fear memory expression, we re-exposed mice to the conditioning context at day 4 or 30 after training while suppressing CFC-tagged hM4Di+ neurons (Fig. 3a, c). Independent groups of mice were used to avoid potentially confounding effects of repeated testing (e.g. extinction) and the possibility of lasting effects of CNO treatment. CNO-induced suppression of CFC-tagged neurons on day 4 (recent memory) did not affect freezing compared with control mice (Fig. 3b). In contrast, at day 30 after training (remote memory), suppression of CFC-tagged mPFC neurons reduced freezing behavior (Fig. 3d). We next examined whether the lack of effect of CNO on recent memory expression could be explained by a difference in the size of the subset of manipulated neurons. To assess this, mice were killed 24 h after the last test. Quantification of the number of hM4Di+ cells in the recent and remote groups revealed no difference (5.7 ± 0.2% and 6.9 ± 0.8%, respectively; Fig. 3e). This, together with the observation that CNO was able to reduce the excitability of hM4Di+ neurons within the first week after CFC (Fig. 2e–g), indicates that potential differences in hM4Di expression likely did not contribute to the differential effect of CNO on recent and remote memory expression. Next, we assessed whether the effect on remote memory could be specifically attributed to CFC-tagged mPFC neurons. To this end, we first exposed mice to a neutral context (context B) and treated animals with 4TM to express hM4Di in mPFC neurons activated by this context (Fig. 3f). Three days later, mice received CFC training in context A and then underwent a remote memory test in context A (day 30 after tagging) in the presence of CNO treatment. Suppression of context B-tagged neurons did not affect freezing in context A (Fig. 3g). Importantly, context B tagged a similar percentage of mPFC neurons (5.9 ± 0.4%) as CFC (compare Fig. 3e, h), indicating that suppression of a different similar-sized subset of mPFC neurons did not affect expression of remote fear memory. Thus, remote, but not recent, memory depended on the activity of mPFC ensembles activated by a single pairing of the CFC context with an aversive stimulus and allocation of fear memory to these specific neurons already occurred during conditioning.

We next determined whether involvement of mPFC neurons in remote memory generalized to stronger fear conditioning. For this, mice received three foot-shocks (3US) during CFC and were treated with 4TM to express hM4Di-mCherry in activated mPFC neurons (Fig. 3i). In contrast with 1US CFC, chemogenetic suppression of tagged mPFC neurons after 3US CFC had no

effect on expression of remote fear memory (Fig. 3j), despite the observation that a similar percentage of mPFC neurons was tagged (Fig. 3k). Given this difference, we investigated whether 1 and 3US CFC evoked differential neuronal activity (Fos expression) in several regions that are known to have a crucial role in contextual fear memory and systems consolidation. Neuronal activity did not differ in the mPFC (PL), posterior ACC and hippocampal subregions (dentate gyrus and CA3), but 3US CFC activated more cells in the basolateral amygdala (BLA) and Reunions thalamic nucleus (Re; Supplementary Fig. 2), in line with the established involvement of these latter regions in remote fear memory when mice are conditioned using multiple foot-shocks[11,26,27].

**Stimulation of CFC-tagged neurons evokes memory expression**. Although mPFC ensembles were not necessary for recent memory expression, we next determined whether chemogenetic stimulation of these neurons is sufficient to evoke fear memory expression at recent and remote time-points after 1US CFC. To test this, mice received AAV-Fos::CreERT2 combined with a Cre-dependent AAV encoding hM3Dq (activating Gq-DREADD[28]) fused to mCherry. Using this virus mixture, we again observed inducible neuronal activity-dependent tagging of mPFC neurons (Supplementary Fig. 3). Next, mice that underwent CFC followed by 4TM treatment were exposed to a neutral context (B) on day 3 and 4 after training and we assessed freezing behavior after vehicle and CNO treatment, respectively (Fig. 4a). In the same mice, we repeated this treatment protocol at day 30 and 31 in a different neutral context (context C). As expected, animals showed minimal freezing in context B and C after vehicle treatment, but CNO enhanced freezing at both time-points (Fig. 4b). Repeated measures ANOVA revealed a significant effect of treatment only, confirming that CNO induced memory expression at both time-points. To verify that CNO increased activity of hM3Dq+ neurons, mice received either vehicle or CNO in their home-cage and were perfused 120 min later to examine colocalization of hM3Dq-mCherry and cells expressing endogenous Fos protein (Fig. 4c, d). Indeed, CNO induced Fos in hM3Dq+ neurons (27 ± 1.3%), whereas very few Fos+ neurons colocalized with hM3Dq+ cells (0.6 ± 0.9%) after vehicle treatment (Fig. 4e). Notably, CNO did not enhance freezing in mice that expressed hM3Dq in mPFC neurons that were activated by exposure to the conditioning context only (in the absence of a foot-shock) or mCherry alone in CFC-tagged mPFC neurons (Supplementary Fig. 4). This indicates that enhanced freezing of the CFC-tagged hM3Dq group was not caused by non-specific effects of CNO, stimulation of a random ensemble in the mPFC, nor by potential formation of an aversive association with the neutral context after vehicle treatment. Taken together, this shows that chemogenetic stimulation of CFC-tagged mPFC neurons was sufficient to induce memory expression at both recent and remote time-points after CFC.

**CFC-tagged neurons are reactivated during remote retrieval**. Although activity of mPFC neurons tagged after 1US CFC was required for remote memory expression only (Fig. 3a–d), chemogenetic stimulation of these cells was sufficient to evoke memory expression already at a recent time-point after CFC. This indicates that mPFC neurons can support recent fear memory. Therefore, we hypothesized that they may not causally contribute to recent memory retrieval, because they are not reactivated upon re-exposure to the conditioning context at this early stage. To study this, we expressed mCherry in neurons that were activated during CFC and then mice were re-exposed to the conditioning context either 4 or 30 days later (Fig. 4f). Ninety minutes after the memory test mice were killed to study expression of Fos (induced by the test) in the mCherry+ and

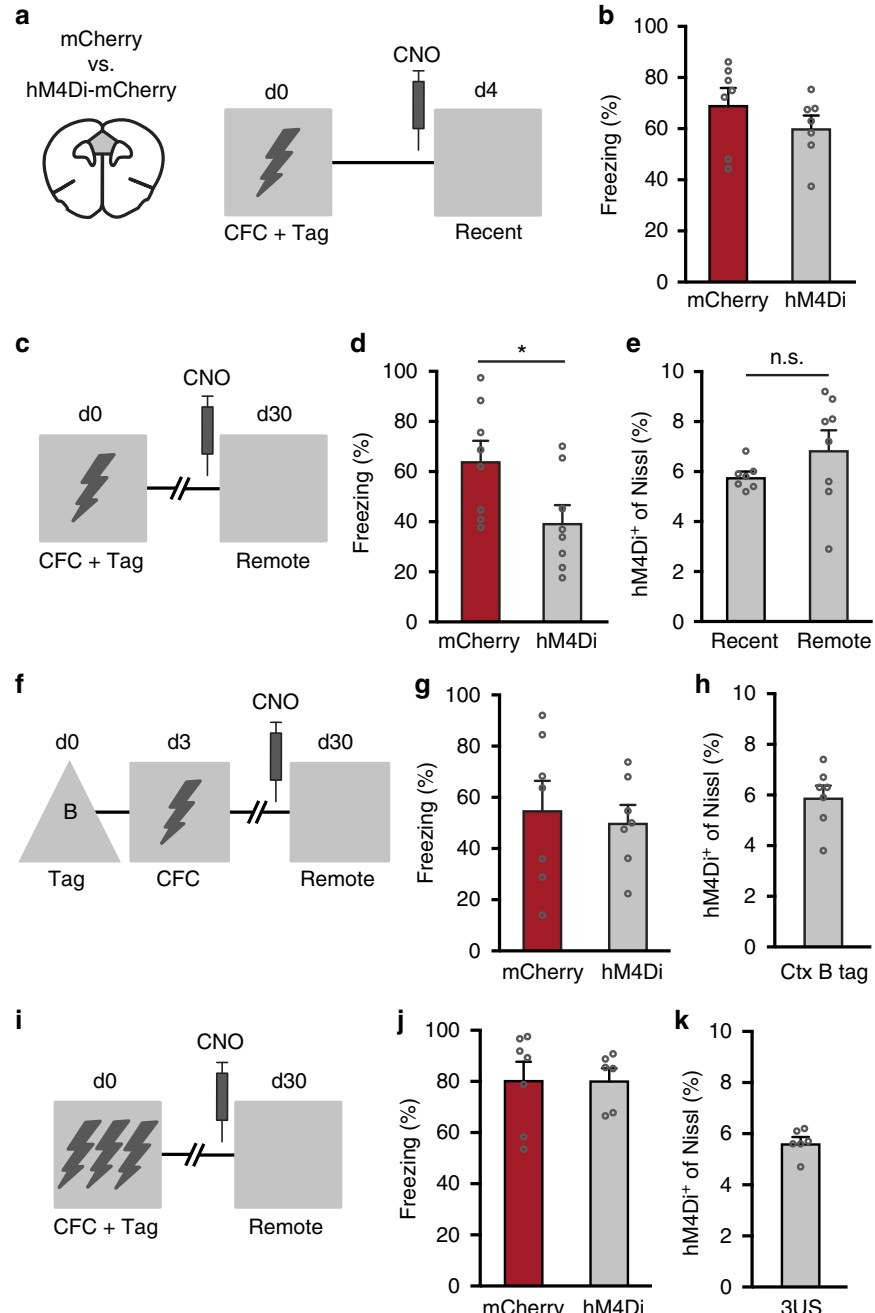

**Fig. 3** CFC-tagged mPFC neurons are selectively involved in remote memory expression. **a** Experimental design. mPFC neurons activated during CFC were tagged with hM4Di-mCherry or mCherry. Both groups received CNO before a recent memory test on day 4. **b** CNO did not affect freezing levels on day 4. Unpaired $t$-test: $t_{12} = 1.169$, $p = 0.265$, mCherry ($n = 7$), hM4Di ($n = 7$). **c** Experimental design. Groups received CNO before a remote memory test on day 30. **d** CNO reduced freezing of the hM4Di group compared with mCherry on day 30. Unpaired $t$-test: $t_{14} = 2.36$, *$p = 0.033$, $n = 8$ per group. **e** Percentage of hM4Di+ neurons in recent and remote groups. Unpaired $t$-test: $t_{13} = 1.413$, $p = 0.18$. n.s. = not significant. **f** Experimental design. mPFC neurons activated by context B were tagged with hM4Di-mCherry or mCherry. Mice received CNO before a remote memory test in the CFC context. **g** CNO did not affect freezing in the CFC context. Unpaired $t$-test: $t_{12} = 0.381$, $p = 0.71$, mCherry ($n = 7$), hM4Di ($n = 7$). **h** Percentage of hM4Di-mCherry+ neurons tagged by context B exposure. **i** mPFC neurons activated during 3US CFC were tagged with hM4Di-mCherry or mCherry. **j** CNO did not affect freezing on day 30. Unpaired $t$-test: $t_{11} = 0.016$, $p = 0.988$, mCherry ($n = 7$), hM4Di ($n = 6$). **k** Percentage of hM4Di+ neurons tagged during 3US CFC. All bar graphs show means+s.e.m. Source data are provided as a Source Data file

mCherry− cell population (Fig. 4g). After recent retrieval, the percentage of Fos+ neurons within the mCherry+ and mCherry− population did not differ, indicating that reactivation of CFC-activated neurons occurred by chance at this time-point. In contrast, mCherry+ neurons showed enhanced reactivation during retrieval of remote fear memory (Fig. 4h). Furthermore, we hypothesized that

the lack of effect of chemogenetic suppression of tagged mPFC neurons after 3US CFC was due to reduced involvement of these cells in remote memory retrieval. Indeed, in mice that underwent 3US CFC, we found that CFC-tagged mPFC neurons were not reactivated above chance level during remote memory retrieval (Fig. 4i–k). Hence, this confirms that CFC-activated mPFC neurons were

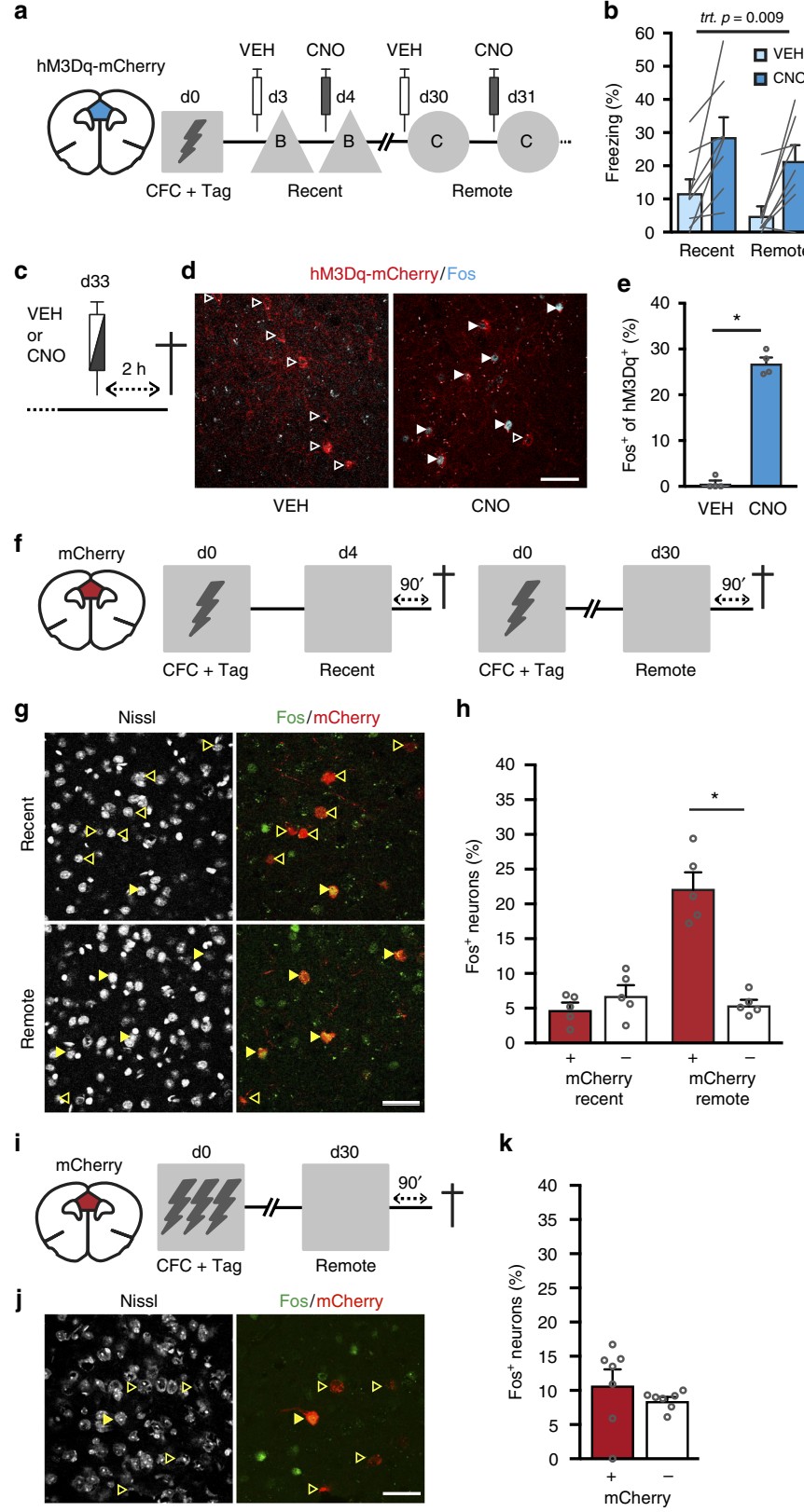

preferentially reactivated during remote, but not recent memory retrieval, and only following 1US conditioning.

**Stimulation of remote memory-tagged neurons evokes freezing.** CFC-activated neurons were not reactivated during recent retrieval, but this did not exclude the possibility that neurons activated during recent retrieval can also mediate expression of fear memory. To functionally investigate this, we determined whether chemogenetic stimulation of mPFC neurons tagged by recent and remote memory retrieval tests could subsequently evoke memory expression in a neutral context (Fig. 5a, c,

**Fig. 4** Stimulation and reactivation of CFC-tagged mPFC neurons. **a** Experimental design. mPFC neurons activated during CFC were tagged with hM3Dq-mCherry. Freezing levels were assessed after vehicle (VEH) and CNO treatment in context B (recent) and C (remote). **b** CNO-enhanced freezing at recent and remote time-points. Repeated measures ANOVA, treatment: $F_{(1,7)} = 13.1$, $p = 0.009$ ($n = 8$ mice). **c** On day 33, mice received VEH or CNO, remained in their home-cage and were perfused 2 h later. **d** hM3Dq-mCherry and Fos expression in mPFC after VEH or CNO treatment. White outlined arrowheads indicate hM3Dq-mCherry+/Fos− cells; white filled arrowheads indicate hM3Dq-mcherry+/Fos+ cells. **e** Percentage of hM3Dq-mcherry+ cells that expressed Fos after VEH or CNO. Mann–Whitney $U = 0$, $p = 0.017$ ($n = 4$ per treatment). **f** Experimental design. mPFC neurons were tagged with mCherry after CFC and re-exposed to the conditioning context 4 or 30 days later. **g** Example of colocalization of mCherry+ and Fos+ cells in the mPFC. Yellow outlined arrowheads indicate mCherry+/Fos− cells; yellow filled arrowheads indicate mCherry+/Fos+ cells. **h** Percentage of Fos+ cells within the mCherry + and mCherry− populations. Two-way repeated measures ANOVA revealed a significant Time-point x Population interaction: $F_{(1,8)} = 93.601$, $p < 0.001$. Post-hoc Bonferroni test: Remote mCherry+ vs. mCherry− *$p < 0.0001$; Recent mCherry+ vs. Remote mCherry+ $p < 0.0001$; Recent mCherry− vs. Remote mCherry+ $p < 0.0001$; $n = 5$ per group. **i** mPFC neurons were tagged with mCherry after 3US CFC and re-exposed to the conditioning context 30 days later. **j** Example of colocalization of mCherry+ and Fos+ cells in the mPFC. Yellow outlined arrowheads indicate mCherry+/Fos− cells; yellow filled arrowheads indicate mCherry+/Fos+ cells. **k** Percentage of Fos+ cells within the mCherry+ and mCherry− populations. Paired $t$-test: $t_6 = 1.186$, $p = 0.281$ ($n = 7$ mice). All bar graphs show means + s.e.m. Scale bars = 50 µm. Source data are provided as a Source Data file

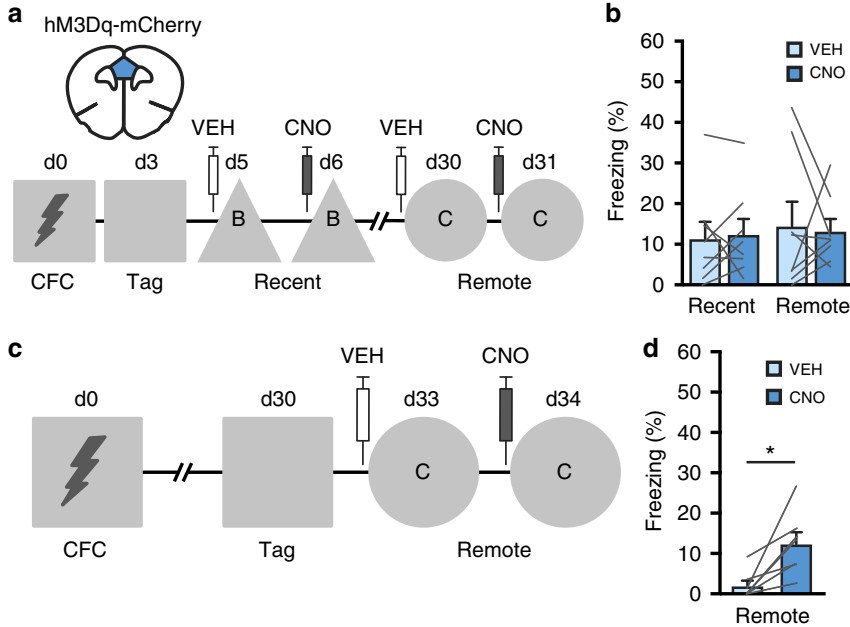

**Fig. 5** Stimulation of retrieval-tagged mPFC neurons. **a** Experimental design. mPFC neurons were tagged with hM3Dq-mCherry after recent retrieval in the CFC context and freezing was subsequently assessed in context B and C after VEH and CNO treatment. **b** Repeated measures ANOVA did not reveal differences in freezing levels between VEH and CNO sessions at both time-points (treatment: $F_{(1,7)} = 0.169$, $p = 0.69$; $n = 8$ mice). **c** Experimental design. mPFC neurons were tagged with hM3Dq-mCherry after remote retrieval in the CFC context and freezing was subsequently assessed in context C. **d** CNO induced freezing compared with VEH. Paired $t$-test, $t_6 = 3.56$, *$p = 0.012$ ($n = 7$ mice). All bar graphs show means + s.e.m. Source data are provided as a Source Data file

respectively). After 1US CFC, mPFC neurons were tagged by re-exposing mice to the CFC context (no shock) on day 3 or 30 after training. CNO-induced stimulation of mPFC neurons tagged with hM3Dq during recent retrieval did not enhance freezing in a neutral context on days 6 and 31 after training (Fig. 5b). In contrast, chemogenetic stimulation of mPFC neurons tagged during remote retrieval (day 30) enhanced freezing on day 34 compared with the vehicle session on day 33 (Fig. 5d). Thus, in contrast to remote memory, mPFC neurons activated during recent memory retrieval likely do not encode conditioned-fear memory. Remarkably, a similar percentage of mPFC neurons was tagged after CFC and recent retrieval (6.9 ± 0.7% and 6.5 ± 0.9%, respectively; Supplementary Fig. 5), but less neurons were tagged after remote retrieval (3.6 ± 0.2%). Thus, despite the observation that during recent retrieval more mPFC neurons were tagged, these cells were not sufficient to enhance freezing behavior, whereas a smaller neuronal subset tagged during remote retrieval was sufficient to at least partially recover memory expression.

**Remote memory depends on CREB function in CFC-tagged neurons.** Lastly, we hypothesized that CREB signaling in mPFC neurons activated during 1US CFC is necessary for systems consolidation and memory persistence. To test this, we generated a Cre-dependent AAV encoding mutant CREB^S133A (AAV-hSyn:: DIO-EGFP-mCREB), a well-established repressor of CREB-mediated gene transcription[17,29]. We first confirmed on day 4 after CFC that expression of mCREB was induced in mPFC neurons and controlled by 4TM (Fig. 6a, b). Notably, we found that mCREB expression was already detectable 24 h after CFC (Supplementary Fig. 6). Next, mCREB or mCherry was expressed in mPFC neurons activated during CFC and fear memory was assessed 4 or 30 days later (Fig. 6c). Expression of mCREB in CFC-tagged mPFC neurons did not alter freezing behavior during a recent memory test (Fig. 6d), but impaired freezing during the remote test (Fig. 6e). Generalization of contextual memory was not induced by mCREB as both groups showed similar low levels of freezing in a neutral context (Supplementary Fig. 7). CFC evoked mCREB expression in 6.8 ± 0.4% of mPFC neurons

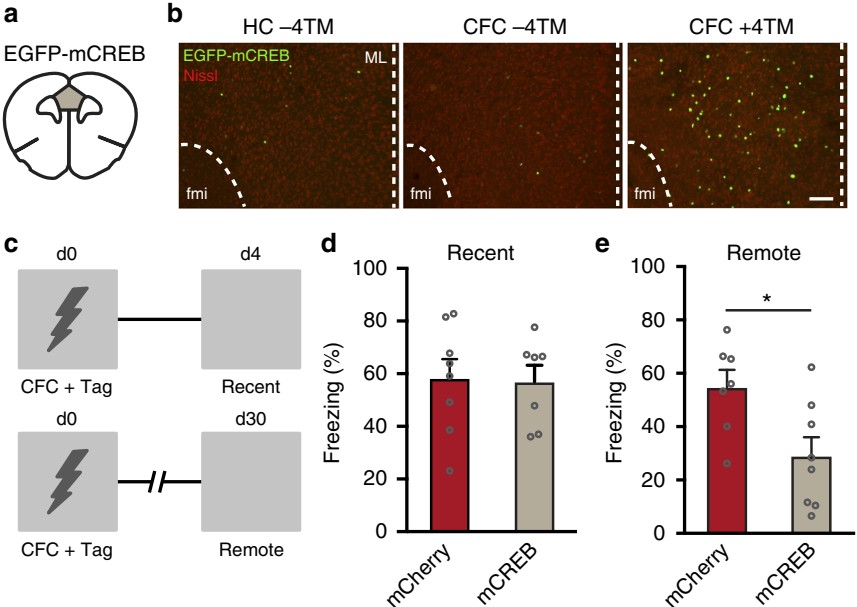

**Fig. 6** Disruption of CREB function in mPFC engram cells impairs remote fear memory. **a** Mice received AAV-Fos::CreER$^{T2}$ combined with AAV-hSyn::DIO-EGFP-mCREB into the mPFC. **b** Representative expression of EGFP-mCREB in mPFC in mice that remained in their home-cage (HC −4TM) and mice that underwent CFC without and with 4TM treatment. Mice were killed 4 days later. Scale bar = 100 μm. fmi = forceps minor of the corpus callosum. ML = midline. **c** Experimental design. mPFC neurons activated during CFC were tagged with EGFP-mCREB and memory was assessed on day 4 and 30 in the conditioning context. **d** On day 4, freezing did not differ between mCREB and control mice. Unpaired $t$-test: $t_{13} = 0.159$, $p = 0.876$. mCherry ($n = 8$), mCREB ($n = 7$). **e** Freezing was significantly reduced in the mCREB mice compared with control mice. Unpaired $t$-test: $t_{13} = 2.669$, *$p = 0.019$. mCherry ($n = 7$), mCREB ($n = 8$). All bar graphs show means + s.e.m. Source data are provided as a Source Data file

(Supplementary Fig. 7c), similar to with what we observed 24 h after CFC (Supplementary Fig. 6) and with hM4Di and hM3Dq (Fig. 2 and Supplementary Fig. 3, respectively). As *Fos* transcription is regulated by phosphorylation of CREB at ser133[30], we hypothesized that mCREB-expressing mPFC neurons should not show Fos induction after a remote memory test. Indeed, we found a complete segregation of mCREB$^+$ and Fos$^+$ cells after a remote test (Supplementary Fig. 8). Hence, disruption of CREB function in mPFC neurons activated during CFC induced a time-dependent impairment in conditioned freezing.

## Discussion

Using a viral-TRAP based approach, we demonstrate that contextual fear memory induced by a single US exposure is allocated to neuronal ensembles in the mPFC during memory encoding and that the activity of these specific neurons is subsequently necessary and sufficient for memory expression one month later. Chemogenetic suppression of a different similar sized "neutral" subset of neurons in the mPFC did not affect remote fear memory, confirming the selectivity of our tag approach and the memory-encoding specificity of activated neuronal ensembles in the mPFC. Strikingly, we found that mPFC ensembles were not involved in remote memory expression after strong (3US) fear conditioning.

Mounting evidence supports the involvement of learning-activated neuronal ensembles in subsequent memory expression[3–6]. For ensembles in the mPFC, this causality appears to be modulated by memory strength. It remains to be determined why mPFC ensembles do not contribute to the expression of a stronger fear memory, but our data is in line with a lack of effect of global prelimbic cortex inactivation on expression of a month-old fear memory when animals are conditioned with multiple foot-shocks[9,26]. Furthermore, we found that, compared with mild conditioning, strong conditioning induced more activated neurons in the BLA and Re, but not in the mPFC. A strong fear memory may therefore involve a relative increase in the contribution of the BLA and Re to systems consolidation. In support of this, the BLA mediates integration of foot-shock and contextual information[31,32] and the Re has been implicated as a critical network hub in remote fear memory when mice received multiple foot-shocks during CFC[26,33]. We compared Fos expression after mild and strong CFC in a number of regions that have previously been implicated in the processing of contextual fear memory. This, however, does not exclude the possibility that neuronal activity in other regions than those we examined is enhanced during strong CFC and that additional regions are engaged in consolidation and retrieval of a strong fear memory[33]. Therefore, we speculate that a strong fearful experience results in recruitment of a more extensive neuronal circuit, with the Re and BLA acting as critical hubs in this engram network. As a result of this broader circuit, the involvement of the mPFC ensemble that is activated during conditioning may be diminished, potentially reflecting a loss of top-down control by the mPFC after a severely aversive experience[34]. In line with this reasoning, it was recently suggested that the mPFC is engaged in the processing of conditioned fear when the threat level is low, but not when it the threat is high[35]. Thus, our study indicates that the strength of an aversive learning experience affects the composition of ensembles that together form a persistent memory engram. Future research should provide insight in the circuitry that gates the recruitment of mPFC neurons that are activated during CFC in expression of remote fear memory.

Our observation that fear-encoding mPFC neurons after mild CFC were not reactivated during recent retrieval of contextual fear memory is in agreement with a recent study[11]. However, this correlative evidence did not address the possibility that neurons activated during recent retrieval can also mediate conditioned freezing behavior. By tagging mPFC neurons that were activated during memory retrieval, we demonstrate that chemogenetic stimulation of neurons activated during remote retrieval can partially recover memory expression, whereas this is not true for neurons activated during recent retrieval. Together, this indicates

that fear-encoding cells in the mPFC are not yet involved in memory retrieval evoked by the conditioning context at recent time-points after learning. Hence, the mPFC engram circuit may initially persist in a dormant state. Why mPFC engram cells are not required for recent memory retrieval remains currently unknown, but they may not be involved yet, because a network of engram cells in other brain regions controls memory expression at this stage. This early network is potentially dominated by the hippocampus, as hippocampal engram cells are critical for recall within the first week(s) after memory acquisition[5,6,11]. Furthermore, systems consolidation by engram cells in cortical regions, such as the mPFC, is thought to require interaction between hippocampus and cortical modules[36,37] and strengthening of connectivity between engram cells in different cortical regions in the first days to weeks after learning[7]. These time-dependent processes may engage cortical engram cells to progressively contribute to memory expression.

Neuronal ensembles activated during mild CFC were preferentially reactivated during remote memory expression. Interestingly, we found that only a subset of the learning-activated neurons were reactivated, in line with previous reports[5,11,38]. This could indicate refinement or contraction of the engram size over time[39]. Alternatively, the partial reactivation of learning-activated cells might be explained by an overestimation of the number of neurons involved in encoding of the memory or by the possibility that memory retrieval only requires reactivation of a subset of the engram population. The latter is in line with our observation that the size of the engram population tagged by remote memory retrieval was smaller than the population tagged by CFC, despite that CFC and remote retrieval activated a similar percentage of neurons in the mPFC (Supplementary Fig. 9). Potentially, expression of a molecular tag using viral-TRAP is only detectable when activation of the Fos promoter exceeds a certain threshold, which may have occurred only in the subset of engram cells that was reactivated during remote retrieval.

Although the precise molecular mechanisms that contribute to maturation of cortical networks supporting fear memory are yet unknown, to our knowledge, we provide the first evidence that CREB signaling in cortical engram cells is crucial for consolidation and subsequent remote memory expression. Our CREB intervention differs from previous studies in important aspects. Firstly, we disrupted CREB function selectively in cortical neurons that were activated during learning, instead of systemically[15], forebrain-wide[17], or non-discriminatively in the majority of neurons of a defined brain region[16]. Secondly, previous reports show that manipulation of CREB function in a subset of neurons prior to fear learning affects the probability that these neurons will participate in encoding of an aversive or appetitive memory[18,19,40]. In contrast, here the CREB repressor was induced after learning, and therefore endogenous selection defined the neurons that encoded the fear memory. The CREB-dependent changes in gene expression that support systems consolidation by engram cells in the mPFC remain to be determined, but they likely involve genes supporting synaptic and structural plasticity processes, as reported for other brain regions[41–44].

To conclude, we demonstrate that upon a mild fearful experience the fear memory is allocated to cortical neurons already during learning and is thus not gradually transferred from the hippocampus to the neocortex after the experience. Together, our data provide crucial insight into the spatiotemporal principles of memory consolidation in cortical networks and reveal that the strength of an aversive learning experience determines whether neuronal ensembles in the mPFC will function as an important network hub in expression of remote memory following a time- and CREB-dependent maturation process.

## Methods

**Animals**. Male wild-type C57BL/6 J mice aged 2–3 months at the start of experiments were individually housed on a 12-h light/dark cycle with food and water available ad libitum. Behavioral experiments were performed during the light phase and mice were randomly assigned to experimental groups. We have complied with all relevant ethical regulations for animal testing and research. All experimental procedures were approved by The Netherlands central committee for animal experiments (CCD) and the animal ethical care committee (DEC) of the Vrije Universiteit Amsterdam.

**Constructs**. The pAAV-Fos::CreER$^{T2}$ plasmid was generated by replacing the coding sequence of tTA in pAAV-cFos−tTA-pA (gift from William Wisden, Addgene plasmid #66794) with the coding sequence of CreER$^{T2}$ from pRetroQ-Cre-ERT2 (gift from Richard Youle, Addgene plasmid #59701) using SLiCE[45]. Similarly, we used SliCE to replacing the coding sequence of mCherry in pAAV-hSyn-DIO-mCherry (gift from Brian Roth, Addgene plasmid #50459) with the sequence of EGFP-mCREB in pAAV-mCREB (gift from Eric Nestler, Addgene plasmid #68551) to produce pAAV-hSyn::DIO-EGFP-mCREB.

**AAV vectors and stereotactic micro-injections**. AAV-Fos::CreER$^{T2}$ (titer: 1.2 × 10$^{13}$) and Cre-dependent AAVs AAV-hSyn::DIO-hM3Dq-mCherry, AAV-hSyn::DIO-hM4Di-mCherry, AAV-hSyn::DIO-mCherry (titers: 5.0–6.0 × 10$^{12}$) and AAV-hSyn::DIO-EGFP-mCREB (titer: 3.0 × 10$^{12}$) were packaged as serotype 5 virus. For stereotactic micro-injections in the mPFC[46], mice first received 0.1 mg per kg Temgesic (RB Pharmaceuticals, UK) and were then anesthetized with iso-flurane and mounted in a stereotactic frame. Lidocaine (2%, Sigma-Aldrich Chemie N.V, The Netherlands) was topically applied to the skull to provide local analgesia. A virus mixture of AAV-Fos::CreER$^{T2}$ and Cre-dependent AAV (ratio 1:500; AAV-Fos::CreER$^{T2}$ was injected at a final titer of 2.4 × 10$^{10}$) was bilaterally injected in the mPFC (+ 1.8 mm AP; ±0.45 mm ML; −2.1 mm DV; relative to Bregma) using microinjection glass needles. Each hemisphere received 0.5 μL of the virus mixture at a flow rate of 0.1 μL per min followed by an additional 5 min to allow diffusion of the virus. Animals remained in their home-cage for 3 weeks until the start of behavioral experiments.

**Contextual fear conditioning**. Mice were first handled for three consecutive days. After an interval of 48 h, mice underwent contextual fear conditioning (CFC)[47,48]. Conditioning was performed in a Plexiglas chamber with a stainless-steel grid floor inside a soundproof cabinet with continuous white noise (68 dB; Ugo Basil, Italy). The CFC context was cleaned with 70% ethanol between each trial. Mice were allowed to explore the CFC context for 120 s prior to the onset of a foot-shock (0.7 mA, 2 s). For 3US conditioning, mice received three foot-shocks (0.7 mA, 2 s) with an interval of 60 s. All mice were returned to their home-cage 30 s after the last foot-shock. Context control groups were allowed to explore the CFC box for 150 s in absence of a foot-shock. Neutral context B (triangular shape, white plastic walls and floor) and C (round shape, white plastic walls and floor) differed in shape and texture and were cleaned with 2% acetic acid. Sessions in context B and C were performed by a different experimenter. During memory tests in context A, B, or C, mice were allowed to explore the context for 2 min. Freezing behavior was analyzed by video tracking using Ethovision XT (Noldus, The Netherlands). Freezing bouts were defined as a lack of movement except respiration for at least 1.5 s.

**4-hydroxytamoxifen treatment**. 4TM (H6278, Sigma-Aldrich Chemie N.V, The Netherlands) was injected in an aqueous solution[25]. First, 15 mg of 4TM was dissolved in 300 μL of DMSO (D8418, Sigma-Aldrich Chemie N.V, The Netherlands). The DMSO stock solution was then diluted in 2850 μl saline containing 2% Tween80 (P1754, Sigma-Aldrich Chemie N.V, The Netherlands) and once more in the same volume of saline. The final solution contained 2.5 mg per ml 4TM, 5% DMSO, and 1% Tween80 in saline. Animals received 4TM (25 mg per kg, i.p.) 2 h after a "tag session" (see experimental design in figures).

**Chemogenetic intervention**. Clozapine N-oxide (CNO; BML-NS105, Enzo Life-Sciences, Brussels) was dissolved in sterile saline. For hM4Di or hMD3q experiments, mice received 5 or 2 mg per kg (i.p.) CNO, respectively, 30 min before a test session.

**Immunohistochemistry**. Mice were transcardially perfused using ice-cold PBS pH 7.4, followed by ice-cold 4% paraformaldehyde (PFA) in PBS pH 7.4. Brains were removed, post-fixed overnight in 4% PFA solution and then immersed in 30% sucrose in PBS with 0.02% NaN$_3$. Brains were then sliced in 35 μm coronal sections using a cryostat and stored in PBS with 0.02% NaN$_3$ at 4 °C until further use. Immunohistochemical stainings were performed using standard procedures[46], using the following antibodies: rabbit anti-Fos (1:500, sc52, Santa Cruz, USA), rabbit anti-RFP (1:1000, Rockland, USA), and NeuroTrace$^{TM}$ 500/525 Green Fluorescent Nissl Stain or NeuroTrace$^{TM}$ 530/615 Red Fluorescent Nissl Stain (1:400, ThermoFisher, USA). Sections were first rinsed in PBS and then incubated with blocking solution containing 5% normal goat serum, 2.5% bovine serum albumin and 0.25% Triton X in PBS at room temperature for 1 h. Primary

antibodies were diluted in blocking solution and sections were incubated with primary antibody solution at 4 °C overnight. Then, sections were rinsed in PBS and incubated with secondary antibodies dissolved in PBS for 2 h at room temperature. NeuroTrace™ for Nissl staining was added to the secondary antibody solution. Finally, sections were rinsed in PBS and mounted using 0.2% gelatin dissolved in PBS. Qualitative expression pictures were generated using a widefield fluorescence microscope (Leica Microsystems, DMi8). For quantification experiments, 6–8 z-stacks per animal were generated using a confocal microscope (Zeiss, LSM510) with the experimenter blinded to the treatment conditions. ImageJ software was used to extract the regions of interest (ROIs) of the cells stained with Nissl (Gaussian filter, Li threshold, watershed). Only ROIs within a predefined range for size (80–2000 square units; to exclude glial cells and non-specific staining) and circularity (0.5 to 1.0) were included. To account for the fact that (parts of the) cells were often present in 2 or 3 images of a z-stack, MATLAB (Mathworks) was used to group the ROIs that belonged to the same Nissl cell and then to count the total number of Nissl$^+$ cells in a z-stack. Cells expressing hM4Di-mCherry, hM3Dq-mCherry, EGFP-mCREB, mCherry or Fos were counted manually.

**Electrophysiological recordings.** Mice were swiftly decapitated and brains were extracted in ice-cold partial sucrose solution (70 mM NaCl, 2.5 mM KCl, 1.25 mM NaH$_2$PO$_4$*H20, 5 mM MgSO$_4$*7H$_2$O, 1 mM CaCl$_2$*2H$_2$O, 70 mM Sucrose, 25 mM D-Glucose, 25 mM NaHCO$_3$, 1 mM Na-Ascorbate, 3 mM Na-Pyruvate, 7.4 pH, 300 mOsm) continuously gassed with carbogen mixture (95% O$_2$, 5% CO$_2$). Acute 300 μm coronal slices containing the mPFC were generated using a vibrating microtome while the brain was submerged in carbogenated ice-cold partial sucrose solution. Slices were transferred in holding ACSF (125 mM NaCl, 3 mM KCl, 1.25 mM NaH$_2$PO$_4$* H$_2$O, 2 mM MgCl$_2$*6H$_2$O, 1.3 mM CaCl$_2$*2H$_2$O, 25 mM D-Glucose, 25 mM NaHCO$_3$, 25 mM D-Glucose, 25 mM NaHCO$_3$, 1 mM Na-Ascorbate, 3 mM Na-Pyruvate, 7.4 pH, 300 mOsm), and left to recover at room temperature for at least 1 h before recording. Subsequently, slices were transferred to a submerged recording chamber, and left to equilibrate for 10 min under continuous perfusion of 2 mL per min of carbogenated running ACSF (= holding ACSF with no Na-Ascorbate or Na Pyruvate and only 1 mM MgCl$_2$*6H$_2$O) supplemented with 10 μM CNQX. The mPFC was identified under visual guidance from differential interference contrast microscopy, and cells expressing either hM4Di-mCherry or mCherry were identified using a mercury-vapor lamp combined with an appropriate fluorescent filter. Whole cell recordings were conducted using borosilicate glass pipettes (2.5–5.5 MΩ) containing K-Gluconate based intracellular (70 mM K-Gluconate, 148 mM KCl, 10 mM Hepes, 4 mM Mg-ATP, 4 mM K2-phospho-creatinine, 0.4 mM GTP, at 280–290 mOsm, 7.2–7.3 pH). Upon establishing a stable giga seal, a step profile was generated from the patched cell, by injecting incrementally increasing current ranging from −100 pA to +300 pA at steps of 10 pA, for 750 ms. Baseline rheobase was assessed by injecting incrementally increasing current ranging from 0 pA up to +400 pA at steps of 20 pA, for 2000 ms. Subsequently, running ACSF containing 50 μm CNO was perfused in, at a rate of approximately 2 mL per min, for at least 5 min, and the ramp and step profile protocols were performed once more. Recordings were acquired with pClamp software (Molecular Devices), using a Multiclamp 700B amplifier (Molecular Devices), sampled at 20 kHz low-pass filtered at 6 kHz, and digitized with an Axon Digidata 1440 A (Molecular Devices).

**Statistical analyses.** Statistical details are presented in the figure legends. Number of animals and number of cells are shown as n. Mice with virus misplacements (in total: hM4Di-mCherry = 6; hM3Dq-mCherry = 5; mCherry = 5; EGFP-mCREB = 1) were excluded from analysis. All graphs show means + s.e.m. SPSS software (IBM) was used for statistical analysis of all data. Comparisons between and within groups were made using two-tailed unpaired or paired Student's t-test, respectively. When the data was not modeled by a normal distribution, it was subjected to non-parametric Mann–Whitney U test for between group comparisons and Wilcoxon signed rank test for within group comparisons. In case of comparisons that involved more than two groups, analyses were performed by One-way ANOVA followed by post-hoc Bonferroni test. In case of more than two within group comparisons, a Repeated measures ANOVA was used. Significance was set at $p < 0.05$.

**Reporting summary.** Further information on research design is available in the Nature Research Reporting Summary linked to this article.

## Data availability
All data generated or analysed during this study are included in this published article (and its Supplementary Information files). Further information and requests for resources and reagents should be directed to Michel C. van den Oever (michel.vanden.oever@vu.nl).

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

## Acknowledgements

We thank Yvonne Gouwenberg for AAV packaging and Lizz Fellinger, Lineke Ouwerkerk, Flora Nelissen and Ioannis Koutlas for assistance with immunohistochemical stainings and imaging. This study was funded by EU MSCA-ITN CognitionNet (FP7-PEOPLE-2013-ITN 607508) to M.R.M., ZonMw VENI grant (916.12.034) to M.C.v.d.O. and VIDI grant (016.168.313) to E.V. and M.C.v.d.O., Amsterdam Neuroscience PoC grant (8-PoC-CIA-2017) to M.C.v.d.O., NWO Veni grant (016.171.033) to P.R-R., and a ZonMw TOP grant (91215030).

## Author contributions

Conceptualization: M.R.M., P.R-R., A.B.S., M.C.v.d.O.; Methodology: M.R.M., R.Z., M.C.v.d.O.; Software: T.G.; Formal analysis: M.R.M., I.K., M.C.v.d.O.; Investigation: M.R.M., E.V., I.K., R.J.v.d.L.; Resources: H.D.M.; Writing – original draft: M.R.M., P.R-R., A.B.S., M.C.v.d.O.; Supervision: A.B.S., M.C.v.d.O.; Funding acquisition: P.R-R., A.B.S., M.C.v.d.O.

## Additional information

**Competing interests:** The authors declare no competing interests.

**Journal Peer Review Information:** *Nature Communications* thanks the anonymous reviewers for their contribution to the peer review of this work. Peer reviewer reports are available.

