## [Peer Review File · Nature Communications]

Reviewers' comments:

Reviewer #1 (Remarks to the Author):

To authors,

Main purpose of this study is to investigate whether the mPFC engram cells for remote memory of contextual fear conditioning is determined at the time of learning or it is formed later on after systems consolidation. To address this important question, the authors used viral vector-mediated TRAP method to selectively label and manipulate the c-Fos activated cells in the mPFC during contextual fear conditioning. By using DREAD system, such labeled mPFC cells were artificially activated or inhibited to determine whether they are engram cells at recent or remote time point. The authors found that those cells were necessary for remote memory retrieval but not for recent memory. Interestingly, activation of those cells in the neutral context was sufficient to induce freezing at both recent and remote time, suggesting that they are already engram cells even at recent time. Finally, the authors used same TRAP strategy to selectively express dominant negative form of CREB in the mPFC cells activated during learning and found that remote memory retrieval is impaired. Together, these results provide an idea that the engram for contextual fear conditioning is formed in the mPFC at the time of learning but they are used for memory retrieval at remote time.

The findings from this study show an important advance in understanding memory engram formation and maturation through systems consolidation. However, due to the recent paper (Kitamura et al., 2017), the novelty of this study is weak.

I have some minor concerns.

1. In a few figures (for instance, Fig. 2b, 2g), I see that individual variation of freezing level in a control group seems to be too big. The authors may need to increase group size.
2. In Fig. 3, the authors claim that stimulating CFC-tagged mPFC neurons evokes memory expression because such manipulation induced freezing. What happens if mPFC neurons tagged with either context (CS) or shock (US) alone is activated?
3. In Fig. 4, the baseline freezing level in the neutral context 'C' in remote test is different in b and d. Because CNO-induced freezing level is similar in both cases, it is necessary to confirm this result in a different condition where baseline freezing level is consistent between groups.
4. In Fig. 5, the authors need to show the time course of CREB activation in the mPFC neurons after contextual fear conditioning and also when mCREB starts to show expression in a TRAP system.

Reviewer #2 (Remarks to the Author):

Time- and CREB-dependent systems consolidation of a cortical fear engram

Mariana R. Matos et al

The goal of this study is to test the role of mPFC, and associated CREB function, in long-term contextual fear memory. The authors used elegant approaches, including DREADDS and a modified TRAP system, to tag and manipulate mPFC neurons involved in encoding and retrieval of contextual fear memory (CFC). The interesting findings presented by the authors are in-line with previous literature that show that the mPFC is needed for retrieval of remote but not recent CFC memory and that a CFC memory established in mPFC during training is need for remote memory. Also, the authors demonstrate a role for CREB in regulating the consolidation of long-term memories in the mPFC.

The authors present a compelling set of results that point to the importance of the mPFC in the

retrieval of remote memories.

Major critique

1. The authors present an exciting novel viral expression system and as such the authors should validate it. For example:

a) They could test co-expression of cFos with CRE and/or hM4Di or hM3Dq, during encoding or retrieval. This analysis is missing throughout the study, making the interpretation of the findings difficult. These data would also estimate the percentage of the engram cells affected by the manipulations. Also, since the virus transfection is likely to affect only the region around the injection site, it is important to determine what percentage of the total mPFC was affected by the manipulation.

b) In figure 1 the authors show that 4 days following the CFC tagging protocol, without any manipulation, 2% of the cells express the inhibitory DREADD hM4Di. Conditioning and CRE activation increase this percentage to 6% and 8% (varies between experiments). This means that approximately 30% of the effect the authors demonstrate is non-specific. Besides referring to this "leaking" in the main text and how it might affect the interpretation of their findings, the authors should characterize it also in the hM3Dq DiO virus. This non-specific expression should be tested for both DREADDs 30 days following virus injection to confirm that it does not increase with time. In addition, for the hM4Di it would be useful to test the freezing levels at day 30 with vehicle prior to CNO to make sure that the lower freezing in the hM4Di group (figure 2d) is directly caused by the DREADD activation.

2. This study utilizes excitatory and inhibitory DREADDs to uncover the functional role of the mPFC in encoding and retrieval of fear memory following short and long intervals. However, several control experiments are missing and there are some discrepancies between the presented experiments that need to be addressed.

a) Comparing freezing levels between day 4 and day 30 (figure 2b+2d), the levels of hM4Di mice are the same between the two days while the levels of the control mice are higher for the long-term retrieval. The authors should explain why the freezing level for the control group is higher, and in light of the above, offer an alternative explanation to their findings.

b) The presented CFC tagging protocol might overestimate the number of neurons involved in the encoding of the memory. Previous literature suggests that while immediately after learning many cells are activated, the retrieval of the memory involves only a subset of these cells. Since the authors tag all the neurons that are activated during learning, some of these neurons might not be crucial for the retrieval of the engram. The authors should refer to this in the interpretation of their findings.

Reviewer #3 (Remarks to the Author):

Matos et al. use a tagging approach based on cFos expression upon learning to access and manipulate "engram neurons" in mPFC (dorsal prefrontal cortex and ACC) induced upon contextual fear conditioning (CFC). They investigate whether these neurons induced at the time of learning might have a role in remote but not recent memory retrieval, a notion central to models of systems consolidation of memories. They find that, indeed, silencing the tagged neurons reduces freezing upon remote but not recent recall. They further tag mPFC neurons upon recent and remote recall and report that only the remote neurons are able to induce some freezing upon reactivation in a neutral context. Finally, they introduce a Cre-dependent mCREB construct in mPFC tagged neurons and show that it interferes with remote recall of fear memory.

The authors conclude that "engram neurons" in mPFC are already established upon fear conditioning but that their (CREB-dependent) role in memory retrieval only becomes essential for remote

memories.

This is a potentially interesting series of experiments addressing issues of general importance concerning long-term consolidation of memories. However, the study in its present form has a number of important weaknesses that substantially diminish its potential significance (and the interpretation of the findings). Important weaknesses involve the distinction between memory strength and "systems consolidation to mPFC", the very modest extent of the fear memory (particularly in the reactivation experiments of Fig4), and the possible interpretation of the CREB interference experiment in Fig. 5.

Specific points:

- 1) A main weakness is that the authors use a very mild CFC protocol (only 1US; only 40% freezing at recent recall) to draw general conclusions about memory consolidation, which might instead relate to time-dependent weakening of memories. They should add experiments with a robust protocol (e.g. 5US) to determine whether they find the same dependency on mPFC tagged cells specifically for remote recall.
- 2) Furthermore, silencing cells tagged in mPFC at remote recall only reduced freezing from 60% to 40% (i.e. to the levels detected at recent recall), indicating that while mPFC cells might have a more detectable impact at remote recall (or might just account for the extra freezing at remote recall), freezing is still robust by the criteria of this study. mPFC "engram neurons" are therefore not essential for remote recall, but at best "involved". This raises the question of what other areas are important. The obvious candidate would be the hippocampus, and the authors might want to compare the impact of hippocampus "engram" silencing at recent and remote recall.
- 3) The experiments in Fig. 4 are difficult to conceptualise. The authors should do similar experiments looking for cFos induction upon recent and remote recall in mPFC and hippocampus (as a control). If, as I suspect, cFos induction in mPFC upon recall will be detectable in both cases, the result is very difficult to explain, particularly in view of the fact that reactivation of the mPFC "engram neurons" tagged upon learning was sufficient to induce freezing in a neutral context at any time. In view of the fact that the freezing induced upon remote recall/reactivation is only 10% I wonder whether this can be considered a fear memory. Again, one way out might be to repeat these experiments with a stronger fear memory (5US CFC).
- 4) The mCREB data are potentially interesting because they suggest that processes occurring several days after learning memory consolidation (it takes several days to achieve robust Cre-dependent expression in these tagging experiments) still depend on functional CREB in "engram neurons". The authors should test recall at 5-8d (still recent but sufficient for mCREB expression), looking for freezing and cFos expression with and without mCREB expression.
- 5) In Fig. 2e/h the authors suggest that closely comparable numbers of neurons are tagged upon CFC (in context A) or in new context exploration without US (in context B; there was almost no induction in home cage controls). This is surprising since context exploration is usually not sufficient to induce robust cFos expression. Could it be that the dual viral vector approach might introduce a bias for weakly cFos expressing cells? As a control to these experiments, the authors might want to look at the actual cFos induction in mPFC in the two experimental settings, and compare the values to those they obtain in their tagging experiments.

Response to the reviewers

The authors thank the editor and reviewers for their positive comments and their insightful suggestions to strengthen the manuscript. Based on the reviewers' comments, we have performed a substantial number of new experiments to further validate our viral-TRAP approach and to alleviate concerns about differences in memory strength at recent and remote time-points. It is important to mention that in all the experiments we performed for the revision, we observed that control mice now froze at equal levels during recent and remote memory tests (both ~60% of the time, see Fig 3b and 6d). Importantly, we once again found no effect of suppression of CFC-tagged neurons during a recent test, thereby replicating our initial findings and ruling out the option that time-dependent differences in memory strength could explain the differential involvement of mPFC engram cells in recent and remote memory. We speculate that previously observed differences in freezing levels were caused by an aberrant batch of animals. Furthermore, our conclusions are supported by a number of additional experiments that we performed for the revised manuscript, which greatly strengthen interpretation of the data. Changes to the main text are highlighted in grey. Please find below a point-by-point rebuttal (in black) to the reviewers comments (in red).

Reviewer #1 (Remarks to the Author):

1. In a few figures (for instance, Fig. 2b, 2g), I see that individual variation of freezing level in a control group seems to be too big. The authors may need to increase group size.

We agree with the reviewer that in certain experiments, variability within groups was bigger than expected. Therefore, we repeated the experiment in Fig. 2b (now Fig 3b). Despite an overall increase in freezing levels of both groups (compared with the original data), we again found no difference in freezing between groups during a recent memory test after CNO treatment. Notably, we also performed a remote test with this group, which similar to Fig 3d, revealed that CNO reduced freezing in the hM4Di group (see graph below; $p = 0.005$). We did not include this graph in the main text, because we cannot exclude the possibility that CNO treatment during the recent memory test induced a carry-over effect (e.g. impaired reconsolidation). Therefore, we think it is more valid to show the data of independent groups in Fig 3b and d.

2. In Fig. 3, the authors claim that stimulating CFC-tagged mPFC neurons evokes memory expression because such manipulation induced freezing. What happens if mPFC neurons tagged with either context (CS) or shock (US) alone is activated?

We thank the reviewer for suggesting this important control experiment. We tagged activated mPFC neurons after exposure to the CFC context alone and then assessed freezing after vehicle and CNO treatment in a different neutral context (Supplementary Fig. 3a and b). We found no significant increase in freezing levels after CNO treatment. This confirms that the increased freezing that we observed in Fig 3b (now Fig 4b) was not caused by potential non-specific effects of CNO treatment or stimulation of a random ensemble in the mPFC. We added this conclusion to the main text (lines 211-216). We combined the mCherry alone control experiment (previously Fig 3f-g) with the context exposure alone experiment in Supplementary Fig 3.

3. In Fig. 4, the baseline freezing level in the neutral context 'C' in remote test is different in b and d. Because CNO-induced freezing level is similar in both cases, it is necessary to confirm this result in a different condition where baseline freezing level is consistent between groups.

Baseline levels are indeed slightly different between these experiments. Although baseline levels can differ slightly between batches of animals, we think that for these particular experiments the enhanced baseline freezing in Fig 5b is caused by a first re-exposure to the conditioning context at a recent time-point after CFC. After a recent memory test, we typically observe higher baseline freezing levels in a neutral context. We repeated the remote retrieval-tag experiment with the expectation that baseline freezing in the vehicle session would increase, but freezing levels were again low and at the level of the vehicle group as shown in Fig. 5d. Based on a suggestion by reviewer 3, we performed CFC using a stronger training protocol (using 3 shocks). With this protocol, we indeed observed slightly higher baseline freezing levels, which were modestly, but significantly increased by CNO treatment (see below).

We did not include this 3US exp in the manuscript, because we think that one of the strengths of our study is the consistent use of a single US learning trial in all experiments.

4. In Fig. 5, the authors need to show the time course of CREB activation in the mPFC neurons after contextual fear conditioning and also when mCREB starts to show expression in a TRAP system.

We appreciate this suggestion by the reviewer and acknowledge that it would be relevant to show CREB activation in mPFC neurons that are activated during CFC. However, we tried to

analyze activation of CREB (measured by the ratio of total CREB vs. phosphorylated CREB (pCREB)), but unfortunately had to conclude that this was not possible. Analysis of CREB activation in a minority (~6-8%) of mPFC cells cannot be assessed by immunoblotting and requires immunohistological stainings. We have performed these stainings using a widely used pCREB antibody (#06-519, Merck Millipore) and previously described staining protocols^{3, 4}, but found that most, if not all, cells in the mPFC express a baseline level of pCREB. Hence, to determine whether CREB activation is enhanced by fear conditioning, one would have to quantify levels of CREB and pCREB in engram cells before and after training. This is not technically feasible, because prior to fear conditioning we cannot determine which mPFC neurons will become engram cells. Therefore, we cannot reliably quantify levels of CREB activation. However, based on the selective effect of mCREB on remote memory expression and the addition of several new control experiments to the revised manuscript (e.g. Fig. 6c, Supplementary Figs 4 and 6), we think our conclusion that CREB-mediated gene transcription within mPFC engram cells has a critical role in systems consolidation of fear memory is valid. With respect to the expression of mCREB using our viral-TRAP system, we now examined expression as early as 24 h after CFC and found a percentage of mCREB⁺ cells (~6%; Supplementary Fig. 5) that was similar to what we had previously observed at a later time-point (lines 298-299; Supplementary Fig. 6)

Reviewer #2 (Remarks to the Author):

1. The authors present an exciting novel viral expression system and as such the authors should validate it. For example:

a) They could test co-expression of cFos with CRE and/or hM4Di or hM3Dq, during encoding or retrieval. This analyses is missing throughout the study, making the interpretation of the findings difficult. These data would also estimate the percentage of the engram cells affected by the manipulations.

We agree with the reviewer that confirmation of co-expression of Fos and CreER^{T2} would further support the validity of the viral-TRAP approach. We performed immunostainings for Fos and Cre at 2, 3 and 5 h after CFC, but at 2 and 3 h we could only detect Fos⁺ cells (with a decline in Fos numbers over-time, as expected), whereas at 5 h, Fos was no longer detectable and we observed only moderate numbers of faint Cre⁺ cells. Based on this, we speculate that:

1) The expression timeframes of endogenous Fos protein and CreER^{T2} from the viral vector most likely do not overlap.

2) The AAV-Fos-CreERT2 vector was used at a titer (final titer: 2.4×10^{10}) that induced CreER^{T2} expression at levels that were too low to detect with a Cre-antibody.

Therefore, we have to conclude that it is not possible to study co-expression of Fos and CreER^{T2}.

Notably, we have added new experiments to our manuscript that further validate the viral-TRAP method. Firstly, we studied endogenous Fos expression in the mPFC after CFC and observed Fos in ~8% of mPFC neurons (new Fig. 1), similar to the percentage of tagged cells with viral-TRAP (Fig. 2d). Secondly, we found that CFC-tagged neurons were preferentially reactivated (Fos⁺) during remote retrieval (new Fig. 4f-h). Together with the ensemble-specific effects of chemogenetic manipulations (Fig. 3, 5 and Supplementary Fig 3b), these new experiments

strongly support our conclusion that we indeed tag activated mPFC neurons that bear the fear engram.

Also, since the virus transfection is likely to affect only the region around the injection site, it is important to determine what percentage of the total mPFC was affected by the manipulation.

It is indeed important to determine what percentage of the total dorsal mPFC region was tagged using viral-TRAP and this is why we always made image z-stacks (blind) along the entire rostral-caudal mPFC axis to obtain an accurate read-out of the percentage of tagged neurons. Using a volume of 0.5 μ l, we found that the virus mixture spreads throughout this entire region. To demonstrate this, we included an overview of the mPFC comprising 6 images that were taken along the rostral-caudal axis (new Supplementary Fig. 1).

b) In figure 1 the authors show that 4 days following the CFC tagging protocol, without any manipulation, 2% of the cells express the inhibitory DREADD hM4Di. Conditioning and CRE activation increase this percentage to 6% and 8% (varies between experiments). This means that approximately 30% of the effect the authors demonstrate is non-specific. Besides referring to this “leaking” in the main text and how it might affect the interpretation of their findings, the authors should characterize it also in the hM3Dq DiO virus.

The reviewer is correct that tagging of ~2% of mPFC cells with hM4Di in the absence of 4TM treatment is a factor that should be taken into consideration, but several experiments rule out that this non-specific tagging influenced the data. Firstly, we show that during a recent test, suppression of CFC-tagged neurons (including the non-specific minority) has no effect on freezing behavior (Fig. 3b). Secondly, suppression of ensembles (~6% of mPFC neurons) that were tagged after exposure to a neutral context did not affect remote fear memory expression (Fig. 3g). Thirdly, chemogenetic stimulation of cells tagged in a neutral context (Supplementary Fig. 3b) or during recent retrieval (Fig. 5b) was not sufficient to enhance freezing.

Based on the reviewer's suggestion, we characterized the tagging of mPFC neurons with hM3Dq (Supplementary Fig. 2; line 197-198) and found a pattern that was very similar to that presented for hM4Di (Fig. 2). With the hM3Dq mixture, we observed that less than 2% of mPFC cells were tagged in absence of 4TM treatment.

This non-specific expression should be tested for both DREADDs 30 days following virus injection to confirm that it does not increase with time.

This is a relevant comment, but in all experiments, we started with the behavioral procedures 3-4 weeks after virus injection. Moreover, we show that mice perfused after a recent and remote test show similar percentages of tagged cells (Fig. 3e), confirming that the tagging of cells did not increase with time. The same was observed for mPFC neurons tagged with EGFP-mCREB (please compare Supplementary Fig 5c and 6c)

In addition, for the hM4Di it would be useful to test the freezing levels at day 30 with vehicle prior to CNO to make sure that the lower freezing in the hM4Di group (figure 2d) is directly caused by the DREADD activation.

We chose not to perform repeated testing (VEH - CNO) in the conditioning context, because we previously observed that mice will freeze less during a second memory test, which is likely induced by a degree of extinction learning. A reduction in freezing would confound interpretation of a potential suppressive effect of CNO. Moreover, we show in Fig. 3g that CNO did not reduce freezing when hM4Di was expressed in a different ensemble. Hence, the suppressive effect that we observed with CFC-tagged ensembles is most likely the result of the combination of CNO treatment and expression of the DREADD in fear-encoding ensembles.

2. This study utilizes excitatory and inhibitory DREADDs to uncover the functional role of the mPFC in encoding and retrieval of fear memory following short and long intervals. However, several control experiments are missing and there are some discrepancies between the presented experiments that need to be addressed.

a) Comparing freezing levels between day 4 and day 30 (figure 2b+2d), the levels of hM4Di mice are the same between the two days while the levels of the control mice are higher for the long-term retrieval. The authors should explain why the freezing level for the control group is higher, and in light of the above, offer an alternative explanation to their findings.

As mentioned above, we repeated the experiment in Fig 2b (now Fig. 3b) and for control mice, we no longer observed a difference between the recent and remote test (Fig. 3d). Similarly, we also did not observe a difference in freezing of controls groups during a recent and remote test in our mCREB experiments (Fig. 6). We think the previously observed difference might have been caused by batch-to-batch differences in freezing levels. Importantly, the new data does not change interpretation of our findings, but actually strengthens it, as it shows that different freezing levels of control groups cannot explain the differential effect of CNO in hM4Di mice at the recent and remote test.

b) The presented CFC tagging protocol might overestimate the number of neurons involved in the encoding of the memory. Previous literature suggests that while immediately after learning many cells are activated, the retrieval of the memory involves only a subset of these cells. Since the authors tag all the neurons that are activated during learning, some of these neuron might not be crucial for the retrieval of the engram. The authors should refer to this in the interpretation of their findings.

We fully agree with the reviewer that during CFC, more cells may be activated than required for retrieval of the memory. To address this, we examined whether CFC-tagged cells are reactivated during a retrieval test and found that during remote, but not recent, memory retrieval, tagged cells are preferentially reactivated (Fig. 4f-h). In line with the reviewers comment, we found that a subset (~22%) of the tagged cells were reactivated during a remote test, which suggests that only a subset is required for adequate memory retrieval. In addition to this new figure, we included a discussion of this important point on lines 363-374.

Reviewer #3 (Remarks to the Author):

1) A main weakness is that the authors use a very mild CFC protocol (only 1US; only 40% freezing at recent recall) to draw general conclusions about memory consolidation, which might

instead relate to time-dependent weakening of memories. They should add experiments with a robust protocol (e.g. 5US) to determine whether they find the same dependency on mPFC tagged cells specifically for remote recall.

We think we have addressed this concern by replicating Fig. 3b. After repeating this experiment, we did not measure a difference in memory expression at recent and remote time-points in control groups anymore (see also comments above). Both control groups now froze ~60% of the test time. Moreover, we have no evidence that supports a weakening of the memory over-time, as in our previous version of the manuscript, the remote control group even showed higher freezing levels than the recent control group. We think that the consistent use of a single aversive stimulus (US) in all experiments is a strength of our study. Although we cannot exclude that remote memory may become less dependent on mPFC engram cells after training with a stronger training protocol, this does not reduce the significance of our findings for a single associative learning event. We have now mentioned that our results apply to a single learning event in the results section (line 173) and discussion (line 333).

2) Furthermore, silencing cells tagged in mPFC at remote recall only reduced freezing from 60% to 40% (i.e. to the levels detected at recent recall), indicating that while mPFC cells might have a more detectable impact at remote recall (or might just account for the extra freezing at remote recall), freezing is still robust by the criteria of this study. mPFC "engram neurons" are therefore not essential for remote recall, but at best "involved". This raises the question of what other areas are important. The obvious candidate would be the hippocampus, and the authors might want to compare the impact of hippocampus "engram" silencing at recent and remote recall.

We thank the reviewer for this important observation, but our new data (Fig. 3b) rules out that differences in freezing at recent and remote tests affect interpretation of the data. Importantly, previous studies show that even global inactivation of cortical regions only partially attenuates freezing during a remote test^{5, 6}, which is in line with the theory that systems consolidation occurs in multiple cortical modules that may function in parallel⁷. Nonetheless, we agree that 'essential' is an overstatement and replaced 'essential' by 'involved' (line 147 and 177) or 'important' (line 395).

The question of what other regions are important for systems consolidation is highly relevant, but we think that this question is beyond the scope of our study and will not affect our conclusions related to the time-dependent involvement of the mPFC engram in systems consolidation.

3) The experiments in Fig. 4 are difficult to conceptualise. The authors should do similar experiments looking for cFos induction upon recent and remote recall in mPFC and hippocampus (as a control). If, as I suspect, cFos induction in mPFC upon recall will be detectable in both cases, the result is very difficult to explain, particularly in view of the fact that reactivation of the mPFC "engram neurons" tagged upon learning was sufficient to induce freezing in a neutral context at any time.

We agree with the reviewer that for interpretation of the data in Fig. 4 (now Fig. 5), it was important to determine whether mPFC engram cells are reactivated during recent and remote retrieval. We performed this experiment (Fig. 4f-h) and found that compared with non-tagged

mPFC neurons, CFC-tagged neurons are preferentially reactivated during remote retrieval only. Hence, this data supports the chemogenetic stimulation effects presented in Fig 5. Again, we have not examined the hippocampus, because we feel that this is beyond the focus of our manuscript. Moreover, reactivation of engram cells in the hippocampus at recent and remote time-points has been extensively studied by others^{8, 9, 10}.

In view of the fact that the freezing induced upon remote recall/reactivation is only 10% I wonder whether this can be considered a fear memory. Again, one way out might be to repeat these experiments with a stronger fear memory (5US CFC).

We agree that freezing induced by CNO after cells were tagged during a remote test is relatively modest, but want to stress that it is not uncommon to achieve only partial recovery of fear memory (in the range of 10-15% freezing) after artificial stimulation of engram cells^{8, 11}. This might be due to stimulation of only a part of the tagged ensemble (Fig. 4e) or the intensity of stimulation². Also it is likely that artificial stimulation does not capture endogenous firing patterns of engram cells during natural memory retrieval, as was previously discussed by others¹². However, as suggested by the reviewer, we repeated this experiment using a stronger training protocol (3US; 3 foot-shocks is the maximum number of shocks approved by our ethical animal care committee; see figure above). We observed slightly higher freezing levels compared with 1US and a significant effect of CNO, but the difference was again modest. Nonetheless, we think this reflects partial recovery of the fear memory, as we did not detect a significant effect of CNO in several control groups (Supplementary Fig. 3), nor in mice tagged after a recent retrieval test (Fig. 5b), despite similar baseline freezing levels.

4) The mCREB data are potentially interesting because they suggest that processes occurring several days after learning memory consolidation (it takes several days to achieve robust Cre-dependent expression in these tagging experiments) still depend on functional CREB in "engram neurons". The authors should test recall at 5-8d (still recent but sufficient for mCREB expression), looking for freezing and cFos expression with and without mCREB expression.

We thank the reviewer for this insightful suggestion and performed this experiment (Fig. 6c-d). Recent memory expression was not affected by mCREB expression in CFC-tagged mPFC neurons, consistent with our conclusion that these cells undergo a time-dependent process of systems consolidation before they contribute to memory expression.

Based on our new data that mPFC engram cells are not preferentially reactivated during a recent memory test (Fig 4f-h), we examined Fos expression in mCREB-expressing cells after a remote test (which should reactivate engram cells) instead of a recent test. In line with the effect of mCREB on systems consolidation by these cells and regulation of Fos transcription by CREB¹³, we found no overlap of Fos⁺ cells and mCREB⁺ cells (Supplementary Fig. 7; lines 307-310). Hence, this supports our conclusions and confirms that mCREB repressed CREB-mediated gene transcription.

5) In Fig. 2e/h the authors suggest that closely comparable numbers of neurons are tagged upon CFC (in context A) or in new context exploration without US (in context B; there was almost no induction in home cage controls). This is surprising since context exploration is usually not sufficient to induce robust cFos expression. Could it be that the dual viral vector approach might introduce a bias for weakly cFos expressing cells? As a control to these

experiments, the authors might want to look at the actual cFos induction in mPFC in the two experimental settings, and compare the values to those they obtain in their tagging experiments.

We performed this experiment and found that CFC and exploration in a novel context evoked Fos expression in a similar percentage of mPFC neurons (8.5% vs 7.3%, respectively; Fig. 1; lines 79-90), which resembled the percentage of tagged cells using viral TRAP. Thus, we have no indication that viral-TRAP only labels weakly Fos expressing cells. We did detect a small, but significant difference between CFC and novel context exploration, but given the minor difference in percentage of Fos⁺ neurons, we question whether this small difference is functionally relevant. We think that the type of information these activated cells encode is more important, which is underscored by the experiments in Fig. 3, where different ensembles were manipulated.

To conclude, we hope the reviewers share our opinion that the conclusions of our study are supported by the original experiments and comprehensive set of new experiments that we have performed for this manuscript.

References

1. Ryan TJ, Roy DS, Pignatelli M, Arons A, Tonegawa S. Memory. Engram cells retain memory under retrograde amnesia. *Science* **348**, 1007-1013 (2015).
2. Roy DS, Muralidhar S, Smith LM, Tonegawa S. Silent memory engrams as the basis for retrograde amnesia. *Proceedings of the National Academy of Sciences*, (2017).
3. Han JH, *et al.* Neuronal competition and selection during memory formation. *Science* **316**, 457-460 (2007).
4. Zhang H, Kyzar EJ, Bohnsack JP, Kokare DM, Teppen T, Pandey SC. Adolescent alcohol exposure epigenetically regulates CREB signaling in the adult amygdala. *Scientific reports* **8**, 10376 (2018).
5. Frankland PW, Bontempi B, Talton LE, Kaczmarek L, Silva AJ. The involvement of the anterior cingulate cortex in remote contextual fear memory. *Science* **304**, 881-883 (2004).
6. Goshen I, *et al.* Dynamics of retrieval strategies for remote memories. *Cell* **147**, 678-689 (2011).
7. Frankland PW, Bontempi B. The organization of recent and remote memories. *Nat Rev Neurosci* **6**, 119-130 (2005).

8. Kitamura T, *et al.* Engrams and circuits crucial for systems consolidation of a memory. *Science* **356**, 73-78 (2017).
9. Tayler KK, Tanaka KZ, Reijmers LG, Wiltgen BJ. Reactivation of neural ensembles during the retrieval of recent and remote memory. *Curr Biol* **23**, 99-106 (2013).
10. Denny CA, *et al.* Hippocampal memory traces are differentially modulated by experience, time, and adult neurogenesis. *Neuron* **83**, 189-201 (2014).
11. Liu X, *et al.* Optogenetic stimulation of a hippocampal engram activates fear memory recall. *Nature* **484**, 381-385 (2012).
12. Josselyn SA, Kohler S, Frankland PW. Finding the engram. *Nat Rev Neurosci* **16**, 521-534 (2015).
13. Sheng M, McFadden G, Greenberg ME. Membrane depolarization and calcium induce c-fos transcription via phosphorylation of transcription factor CREB. *Neuron* **4**, 571-582 (1990).

Reviewers' comments:

Reviewer #1 (Remarks to the Author):

I do not have any further concerns. I believe that the paper is now acceptable for publication to Nature Communications.

Reviewer #2 (Remarks to the Author):

The authors addressed critical comments of the reviewers with thoughtful responses and most importantly, with additional information. This made a real difference for the presentation of their interesting findings.

Reviewer #3 (Remarks to the Author):

The authors have addressed several of the points raised by the Reviewers, but important issues remain unaddressed (respectively have not been addressed experimentally). From the perspective of this reviewer, that is particularly the case for the issue of how these findings can be generalised to those using more conventional (stronger) conditioning protocols (the very modest extent of remote freezing induced by the tagged neurons is not convincing given the importance of the potential claim). Furthermore, comparison with the "yang" part of these findings (that hippocampal ensembles are not important for remote recall) would have been important to strengthen the notion that mPFC ensembles are specifically important for remote recall of fear memories.

Minor point: given the surprising finding that the magnitude of cFos+ neuron induction in mPFC is almost the same upon context exploration or contextual fear conditioning, it would be informative to know whether the ensembles differ qualitatively, or whether the ensembles in mPFC do not involve a conditioning-specific component (other than how they might be linked to conditioning ensembles in other brain areas).

Response to the reviewers

Reviewer #1

I do not have any further concerns. I believe that the paper is now acceptable for publication to Nature Communications.

We thank the reviewer for the recommendation to accept our manuscript for publication in Nature Communications.

Reviewer #2:

The authors addressed critical comments of the reviewers with thoughtful responses and most importantly, with additional information. This made a real difference for the presentation of their interesting findings.

We thank the reviewer for the positive comments on our responses and the findings presented in our manuscript.

Reviewer #3:

The authors have addressed several of the points raised by the Reviewers, but important issues remain unaddressed (respectively have not been addressed experimentally). From the perspective of this reviewer, that is particularly the case for the issue of how these findings can be generalised to those using more conventional (stronger) conditioning protocols.

To specifically address this point, we have performed several new experiments in which we conditioned mice with a stronger conditioning protocol. As mentioned previously, our animal ethical care committee does not allow the use of 5 foot-shocks and therefore we have used 3 foot-shocks of 0.7 mA (3US CFC). This strong conditioning protocol produced freezing levels of ~80% during remote memory tests (new Fig. 3i, j).

Similar to 1US CFC, we determined whether chemogenetic suppression of mPFC neurons tagged with hM4Di-mCherry after 3US CFC would impair remote memory expression. Strikingly, suppression of mPFC ensembles did not affect remote freezing levels in mice that experienced 3US CFC (new Fig. 3i, j), despite the observation that we tagged a similar number of mPFC neurons with 3US CFC (new Fig. 3k) and 1US CFC. This result is in line with previous studies showing that global inactivation of the prelimbic cortex does not affect systems consolidation and expression of remote fear memory when mice were conditioned with three or more foot-shocks of at least 0.7 mA^{1, 2}.

To further explain this finding, we investigated whether mild (1US) and strong (3US) CFC evoked differential neuronal activity in the mPFC and other regions that have previously been implicated in systems consolidation of contextual fear memory. For this, we compared Fos expression in the prelimbic cortex (PL), posterior anterior cingulate cortex (pACC), dentate gyrus (DG), CA3, basolateral amygdala (BLA) and Reunions thalamic nucleus (Re) after 1 and 3US CFC. We found no difference in the number of Fos⁺ neurons in the PL (in line with the

similar number of tagged neurons in Fig. 3e and k), pACC, DG and CA3, however the BLA and Re showed a significant increase in Fos⁺ neurons after 3US CFC (new Supplementary Figure 2). These findings indicate that a stronger conditioning protocol indeed evokes additional neuronal activity in regions other than the mPFC. Notably, the BLA and Re are well known to function as a critical network hub in systems consolidation of contextual fear memory when animal are conditioned using multiple foot-shocks^{1,3,4}.

Finally, we studied whether chemogenetic suppression of tagged mPFC ensembles had no effect on remote fear memory in mice that underwent 3US CFC (Fig. 3j), because these cells were not reactivated during retrieval. For this, we examined colocalization of Fos induced by a remote memory test and mCherry in mPFC neurons tagged after 3US CFC (new Fig. 4i, j). In contrast with 1US CFC and in support of our hypothesis, we found that tagged mPFC neurons were not preferentially reactivated during remote memory retrieval after 3US CFC (new Fig. 4k). Taken together, our data indicate that a mild fearful experience is consolidated by mPFC neurons (in a CREB-dependent manner) and then requires activity of these same cells for remote retrieval, whereas memory retrieval after a strong fearful experience does not depend on mPFC ensembles. A more severe aversive experience likely engages different or additional neuronal network elements to support systems consolidation. Therefore, we conclude that the strength of fear conditioning determines whether mPFC ensembles functionally contribute to expression of remote fear memory. This is discussed on lines 361-377. To the best of our knowledge, we show for the first time that memory strength gates the involvement of neuronal ensembles that are activated during conditioning in subsequent memory expression. To highlight this finding, we have adjusted the title of our manuscript to "Systems consolidation of a prefrontal cortex fear engram depends on memory strength and CREB function". We thank the reviewer for stressing the importance of investigating whether our initial findings generalized to a stronger fear conditioning protocol, which sparked the generation of this new data set.

Furthermore, comparison with the "yang" part of these findings (that hippocampal ensembles are not important for remote recall) would have been important to strengthen the notion that mPFC ensembles are specifically important for remote recall of fear memories.

We like to explain that it was never our intention to suggest that hippocampal regions are not involved in remote recall. We mentioned that under certain conditions remote fear memory can be retrieved independent of the hippocampus, however, this has been observed only after lesions or prolonged inactivation of the hippocampus, which does not rule out the possibility that hippocampal neurons are involved under endogenous conditions. In fact, the latter is supported by several other studies^{5, 6}. Therefore, we favor the multiple trace theory of systems consolidation, which entails interaction between hippocampal and cortical regions during early and remote stages of memory processing. Importantly, the relative contribution of cortical brain regions to memory retrieval seems to increase in a time-dependent manner, which is also supported by our data when we trained mice with 1US CFC. Therefore the primary focus of our manuscript is on the role of mPFC neuronal ensembles in systems consolidation and remote memory expression. To avoid the confusion regarding the role of the hippocampus, we removed sentences in the introduction and discussion referring to hippocampus-independent retrieval of fear memory.

Minor point: given the surprising finding that the magnitude of cFos⁺ neuron induction in mPFC is almost the same upon context exploration or contextual fear conditioning, it would be

informative to know whether the ensembles differ qualitatively, or whether the ensembles in mPFC do not involve a conditioning-specific component (other than how they might be linked to conditioning ensembles in other brain areas).

We agree that it would be informative to understand whether ensembles activated upon context exploration and contextual fear conditioning differ qualitatively, but as the answer to this question will not affect our conclusions and will require a substantial amount of additional experimental data, we think answering this question is beyond the current scope of study.

References:

1. Vetere G, *et al.* Chemogenetic Interrogation of a Brain-wide Fear Memory Network in Mice. *Neuron* **94**, 363-374 e364 (2017).
2. Frankland PW, Bontempi B, Talton LE, Kaczmarek L, Silva AJ. The involvement of the anterior cingulate cortex in remote contextual fear memory. *Science* **304**, 881-883 (2004).
3. Kitamura T, *et al.* Engrams and circuits crucial for systems consolidation of a memory. *Science* **356**, 73-78 (2017).
4. Silva BA, Burns AM, Graff J. A cFos activation map of remote fear memory attenuation. *Psychopharmacology (Berl)* **236**, 369-381 (2019).
5. Khalaf O, Resch S, Dixsaut L, Gorden V, Glauser L, Gräff J. Reactivation of recall-induced neurons contributes to remote fear memory attenuation. *Science* **360**, 1239-1242 (2018).
6. Goshen I, *et al.* Dynamics of retrieval strategies for remote memories. *Cell* **147**, 678-689 (2011).

REVIEWERS' COMMENTS:

Reviewer #3 (Remarks to the Author):

The authors have now included experiments using a more conventional (stronger) conditioning protocol involving 3US instead of only 1US. The result is very different, as the cFos+ cells in mPFC are now not anymore necessary for remote fear memory retrieval.

The results are now more complete and also more interesting. However, the authors only discuss relatively superficially what might account for the difference between 1US and 3US retrieval. They speculate that additional areas such as BLA, which exhibit more cFos+ neurons upon 3US might have a key role in remote retrieval of stronger fear memories. That might be, but why does PFC become critical for weak memories? Even if the answer is currently unclear/speculative, more discussion of this interesting aspect would strengthen the paper.

Further points:

- 1) The title now says that "Systems consolidation of a prefrontal cortex fear engram depends on memory strength and CREB function", but it seems to me that this is not something the authors have demonstrated. Instead, what they show is that the role of the cFos+ cells in mPFC for remote memory is only essential when fear memories are weak (nothing about systems consolidation).
- 2) It would be important to know whether activation of the 3US mPFC "engram" is also sufficient for fear memory expression (Fig4).
- 3) The summary mentions systems consolidation in mPFC as a function of 1US versus 3US, but what the authors show is dependency on activity in cFos+ cells, not "systems consolidation" (how would one measure that?). Furthermore, any hypothesis as to how mPFC dependency changes between 1US and 3US would also belong in the summary.

Response to the reviewer

The authors want to thank the reviewer for the positive feedback.

Please find below a point-by-point reply (blue) to the comments of the reviewer (black).

Reviewer #3 (Remarks to the Author):

The authors have now included experiments using a more conventional (stronger) conditioning protocol involving 3US instead of only 1US. The result is very different, as the cFos+ cells in mPFC are now not anymore necessary for remote fear memory retrieval.

The results are now more complete and also more interesting. However, the authors only discuss relatively superficially what might account for the difference between 1US and 3US retrieval. They speculate that additional areas such as BLA, which exhibit more cFos+ neurons upon 3US might have a key role in remote retrieval of stronger fear memories. That might be, but why does PFC become critical for weak memories? Even if the answer is currently unclear/speculative, more discussion of this interesting aspect would strengthen the paper.

We can indeed only speculate about this, but we now included a discussion about the mechanism that might underlie the loss of involvement of the mPFC ensemble in consolidation and expression of a strong remote fear memory (lines 378-498). Briefly, we hypothesize that strong CFC recruits a more extensive neuronal engram circuit, with the BLA and Re acting as critical hubs in this circuit. This broader engram circuit may diminish the involvement of the mPFC ensemble that is activated during strong CFC.

Further points:

1) The title now says that "Systems consolidation of a prefrontal cortex fear engram depends on memory strength and CREB function", but it seems to me that this is not something the authors have demonstrated. Instead, what they show is that the role of the cFos+ cells in mPFC for remote memory is only essential when fear memories are weak (nothing about systems consolidation).

We show that mPFC engram cells undergo a time-dependent maturation process before they contribute to remote memory, at a time-scale that matches with systems consolidation rather than synaptic consolidation. However, we have not investigated the process of a network-wide reorganization of memory, another feature of systems consolidation. Therefore, we understand the argument of the reviewer and have changed the title to "Memory strength gates the involvement of a CREB-dependent cortical fear engram in remote memory". This title reflects the main findings of our study more precisely.

2) It would be important to know whether activation of the 3US mPFC "engram" is also sufficient for fear memory expression (Fig4).

This is an interesting suggestion, but the outcome of this experiment would not affect our main conclusions. Therefore, we think this experiment is beyond the current scope of our study.

3) The summary mentions systems consolidation in mPFC as a function of 1US versus 3US, but what the authors show is dependency on activity in cFos+ cells, not "systems consolidation" (how would one measure that?). Furthermore, any hypothesis as to how mPFC dependency changes between 1US and 3US would also belong in the summary.

As mentioned above, we understand this comment of the reviewer and have toned-down conclusions about systems consolidation in the summary. Given the word limit of the summary and the fact that any hypothesis related to the difference in dependency of mPFC ensembles after 1US and 3US is highly speculative, we think these hypotheses are more suitable for the general discussion than the summary. Therefore, our hypothesis related to the difference in mPFC dependency after 1US and 3US CFC is now reflected in the text that we have added to the discussion (lines 378-498).